# Interstellar: Searching Recurrent Architecture for Knowledge Graph Embedding

**Yongqi Zhang**[1,3]    **Quanming Yao**[1,2]    **Lei Chen**[3]

[1]4Paradigm Inc.

[2]Department of Electronic Engineering, Tsinghua University

[3]Department of Computer Science and Engineering, HKUST

{zhangyongqi,yaoquanming}@4paradigm.com, leichen@cse.ust.hk

## Abstract

Knowledge graph (KG) embedding is well-known in learning representations of KGs. Many models have been proposed to learn the interactions between entities and relations of the triplets. However, long-term information among multiple triplets is also important to KG. In this work, based on the relational paths, which are composed of a sequence of triplets, we define the Interstellar as a recurrent neural architecture search problem for the short-term and long-term information along the paths. First, we analyze the difficulty of using a unified model to work as the Interstellar. Then, we propose to search for recurrent architecture as the Interstellar for different KG tasks. A case study on synthetic data illustrates the importance of the defined search problem. Experiments on real datasets demonstrate the effectiveness of the searched models and the efficiency of the proposed hybrid-search algorithm. [1]

## 1 Introduction

Knowledge Graph (KG) [3, 43, 52] is a special kind of graph with many relational facts. It has inspired many knowledge-driven applications, such as question answering [30, 37], medical diagnosis [60], and recommendation [28]. An example of the KG is in Figure 1(a). Each relational fact in KG is represented as a triplet in the form of *(subject entity, relation, object entity)*, abbreviated as $(s, r, o)$. To learn from the KGs and benefit the downstream tasks, embedding based methods, which learn low-dimensional vector representations of the entities and relations, have recently developed as a promising direction to serve this purpose [7, 18, 45, 52].

Many efforts have been made on modeling the plausibility of triplets $(s, r, o)$s through learning embeddings. Representative works are triplet-based models, such as TransE [7], ComplEx [49], ConvE [13], RotatE [44], AutoSF [61], which define different embedding spaces and learn on single triplet $(s, r, o)$. Even though these models perform well in capturing short-term semantic information inside the triplets in KG, they still cannot capture the information among multiple triplets.

In order to better capture the complex information in KGs, the relational path is introduced as a promising format to learn composition of relations [19, 27, 38] and long-term dependency of triplets [26, 11, 18, 51]. As in Figure 1(b), a relational path is defined as a set of $L$ triplets $(s_1, r_1, o_1), (s_2, r_2, o_2), \ldots, (s_L, r_L, o_L)$, which are connected head-to-tail in sequence, i.e. $o_i = s_{i+1}, \forall i = 1 \ldots L - 1$. The paths not only preserve every single triplet but also can capture the dependency among a sequence of triplets. Based on the relational paths, the triplet-based models can be compatible by working on each triplet $(s_i, r_i, o_i)$ separately. TransE-Comp [19] and PTransE [27] learn the composition relations on the relational paths. To capture the long-term information

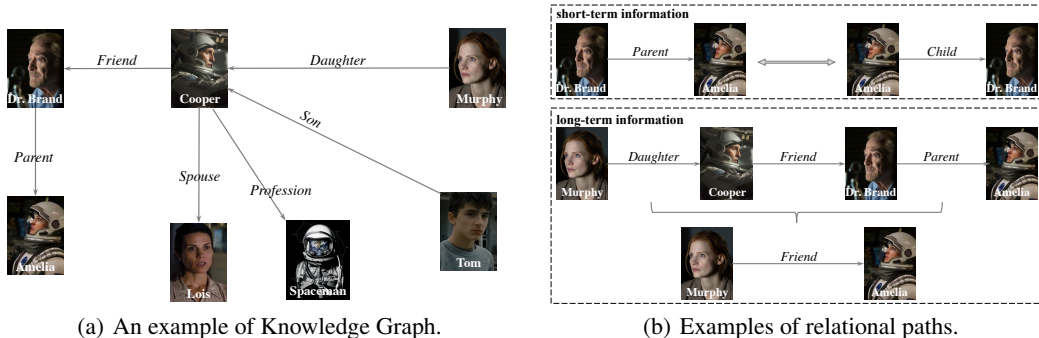

(a) An example of Knowledge Graph.　　　　　　　(b) Examples of relational paths.

Figure 1: Short-term information is represented by a single triplet. Long-term information passes across multiple triplets. The two kinds of information in KGs can be preserved in the relational path.

in KGs, Chains [11] and RSN [18] design customized RNN to leverage all the entities and relations along path. However, the RNN models still overlook the semantics inside each triplet [18]. Another type of models leverage Graph Convolution Network (GCN) [24] to extract structural information in KGs, e.g. R-GCN, GCN-Align [53], CompGCN [50]. However, GCN-based methods do not scale well since the entire KG needs to be processed and it has large sample complexity [15].

In this paper, we observe that the relational path is an important and effective data structure that can preserve both short-term and long-term information in KG. Since the semantic patterns and the graph structures in KGs are diverse [52], how to leverage the short-term and long-term information for a specific KG task is non-trivial. Inspired by the success of neural architecture search (NAS) [14], we propose to search recurrent architectures as the *Interstellar* to learn from the relational path. The contributions of our work are summarized as follows:

1. We analyze the difficulty and importance of using the relational path to learn the short-term and long-term information in KGs. Based on the analysis, we define the Interstellar as a recurrent network to process the information along the relational path.
2. We formulate the above problem as a NAS problem and propose a domain-specific search space. Different from searching RNN cells, the recurrent network in our space is specifically designed for KG tasks and covers many human-designed embedding models.
3. We identify the problems of adopting stand-alone and one-shot search algorithms for our search space. This motivates us to design a hybrid-search algorithm to search efficiently.
4. We use a case study on the synthetic data set to show the reasonableness of our search space. Empirical experiments on entity alignment and link prediction tasks demonstrate the effectiveness of the searched models and the efficiency of the search algorithm.

**Notations.** We denote vectors by lowercase boldface, and matrix by uppercase boldface. A KG $\mathcal{G} = (\mathcal{E}, \mathcal{R}, \mathcal{S})$ is defined by the set of entities $\mathcal{E}$, relations $\mathcal{R}$ and triplets $\mathcal{S}$. A triplet $(s, r, o) \in \mathcal{S}$ represents a relation $r$ that links from the subject entity $s$ to the object entity $o$. The embeddings in this paper are denoted as boldface letters of indexes, e.g. $\mathbf{s}, \mathbf{r}, \mathbf{o}$ are embeddings of $s, r, o$. "$\odot$" is the element-wise multiply and "$\otimes$" is the Hermitian product [49] in complex space.

## 2　Related Works

### 2.1　Representation Learning in Knowledge Graph (KG)

Given a single triplet $(s, r, o)$, TransE [7] models the relation $r$ as a translation vector from subject entity $s$ to object entity $o$, i.e., the embeddings satisfy $\mathbf{s} + \mathbf{r} \approx \mathbf{o}$. The following works DistMult [55], ComplEx [49], ConvE [13], RotatE [44], etc., interpret the interactions among embeddings $\mathbf{s}, \mathbf{r}$ and $\mathbf{o}$ in different ways. All of them learn embeddings based on single triplet.

In KGs, a relational path is a sequence of triplets. PTransE [27] and TransE-Comp [19] propose to learn the composition of relations $(r_1, r_2, \ldots, r_n)$. In order to combine more pieces of information in KG, Chains [11] and RSN [18] are proposed to jointly learn the entities and relations along the relational path. With different connections and combinators, these models process short-term and long-term information in different ways.

Table 1: The recurrent function of existing KG embedding models. We represent the triplet/path-based models by Definition 1. $\mathcal{N}(\cdot)$ denotes the neighbors of an entity. $\mathbf{W} \in \mathbb{R}^{d \times d}$'s are different weight matrices. $\sigma$ is a non-linear activation function. "cell" means a RNN cell [12], like GRU [9] LSTM [46] etc. For mini-batch complexity, $m$ is the batch size, $d$ is the embedding dimension.

| type | model | | unit function | complexity |
|---|---|---|---|---|
| triplet-based | TransE [7] | | $\mathbf{v}_t = \mathbf{s}_t + \mathbf{r}_t, \mathbf{h}_t = 0$ | $O(md)$ |
| | ComplEx [49] | | $\mathbf{v}_t = \mathbf{s}_t \otimes \mathbf{r}_t, , \mathbf{h}_t = 0$ | $O(md)$ |
| GCN-based | R-GCN [40] | | $\mathbf{s}_t = \sigma(\mathbf{s}_{t-1} + \sum_{s' \in \mathcal{N}(s)} \mathbf{W}_t^{(r)} \mathbf{s}'_{t-1})$ | $O(|\mathcal{E}||\mathcal{R}|d)$ |
| | GCN-Align [53] | | $\mathbf{s}_t = \sigma(\mathbf{s}_{t-1} + \sum_{s' \in \mathcal{N}(s)} \mathbf{W}_t \mathbf{s}'_{t-1})$ | $O(|\mathcal{E}|d)$ |
| path-based | PTransE [27] | add | $\mathbf{v}_t = \mathbf{h}_t, \mathbf{h}_t = \mathbf{h}_{t-1} + \mathbf{r}_t$ | $O(mLd)$ |
| | | multiply | $\mathbf{v}_t = \mathbf{h}_t, \mathbf{h}_t = \mathbf{h}_{t-1} \odot \mathbf{r}_t$ | $O(mLd)$ |
| | | RNN | $\mathbf{v}_t = \mathbf{h}_t, \mathbf{h}_t = \text{cell}(\mathbf{r}_t, \mathbf{h}_{t-1})$ | $O(mLd^2)$ |
| | Chains [11] | | $\mathbf{v}_t = \mathbf{h}_t, \mathbf{h}_t = \text{cell}(\mathbf{s}_t, \mathbf{r}_t, \mathbf{h}_{t-1})$ | $O(mLd^2)$ |
| | RSN [18] | | $\mathbf{v}_t = \mathbf{W}_1 \mathbf{s}_t + \mathbf{W}_2 \mathbf{h}_t, \mathbf{h}_t = \text{cell}(\mathbf{r}_t, \text{cell}(\mathbf{s}_t, \mathbf{h}_{t-1}))$ | $O(mLd^2)$ |
| | Interstellar | | a searched recurrent network | $O(mLd^2)$ |

Graph convolutional network (GCN) [24] have recently been developed as a promising method to learn from graph data. As a special instance of graph, GCN has also been introduced in KG learning, e.g., R-GCN [40], GCN-Align [53], VR-GCN [58] and CompGCN [50]. However, these models are hard to scale well since the whole KG should be loaded. for each training iteration. Besides, GCN has been theoretically proved to have worse generalization guarantee than RNN in sequence learning tasks (Section 5.2 in [15]). Table 1 summarizes above works and compares them with the proposed Interstellar, which is a NAS method customized to path-based KG representation learning.

## 2.2  Neural Architecture Search (NAS)

Searching for better neural networks by NAS techniques have broken through the bottleneck in manual architecture designing [14, 20, 41, 63, 56]. To guarantee effectiveness and efficiency, the first thing we should care about is the search space. It defines what architectures can be represented in principle, like CNN or RNN. In general, the search space should be powerful but also tractable. The space of CNN has been developed from searching the macro architecture [63], to micro cells [29] and further to the larger and sophisticated cells [47]. Many promising architectures have been searched to outperform human-designed CNNs in literature [1, 54]. However, designing the search space for recurrent neural network attracts little attention. The searched architectures mainly focus on cells rather than connections among cells [36, 63].

To search efficiently, evaluation method, which provides feedback signals, and search algorithm, which guides the optimization direction, should be simultaneously considered. There are two important approaches for NAS. 1) Stand-alone methods, i.e. separately training and evaluating each model from scratch, are the most guaranteed way to compare different architectures, whereas very slow. 2) One-shot search methods, e.g., DARTS [29], ASNG [1] and NASP [57], have recently become the most popular approach that can efficiently find good architectures. Different candidates are approximately evaluated in a supernet with parameter-sharing (PS). However, PS is not always reliable, especially in complex search spaces like the macro space of CNNs and RNNs [5, 36, 41].

## 3  The Proposed Method

As in Section 2.1, the relational path is informative in representing knowledge in KGs. Since the path is a sequence of triplets with varied length, it is intuitive to use Recurrent Neural Network (RNN), which is known to have a universal approximation ability [39], to model the path as language models [46]. While RNN can capture the long-term information along steps [32, 9], it will overlook domain-specific properties like the semantics inside each triplet without a customized architecture [18]. Besides, what kind of information should we leverage varies from tasks [52]. Finally, as proved in [15], RNN has better generalization guarantee than GCN. Thus, to design a proper KG model, we define the path modeling as a NAS problem for RNN here.

### 3.1  Designing a Recurrent Search Space

To start with, we firstly define a general recurrent function (Interstellar) on the relational path.

**Definition 1** (Interstellar). *An Interstellar processes the embeddings of $\mathbf{s}_1, \mathbf{r}_1$ to $\mathbf{s}_L, \mathbf{r}_L$ recurrently. In each recurrent step $t$, the Interstellar combines embeddings of $\mathbf{s}_t$, $\mathbf{r}_t$ and the preceding information $\mathbf{h}_{t-1}$ to get an output $\mathbf{v}_t$. The Interstellar is formulated as a recurrent function*

$$[\mathbf{v}_t, \mathbf{h}_t] = f(\mathbf{s}_t, \mathbf{r}_t, \mathbf{h}_{t-1}), \quad \forall t = 1 \ldots L, \tag{1}$$

*where $\mathbf{h}_t$ is the recurrent hidden state and $\mathbf{h}_0 = \mathbf{s}_1$. The output $\mathbf{v}_t$ is to predict object entity $o_t$.*

In each step $t$, we focus on one triplet $(s_t, r_t, o_t)$. To design the search space $\mathcal{A}$, we need to figure out what are important properties in (1). Since $\mathbf{s}_t$ and $\mathbf{r}_t$ are indexed from different embedding sets, we introduce two operators to make a difference, i.e. $O_s$ for $\mathbf{s}_t$ and $O_r$ for $\mathbf{r}_t$ as in Figure 2. Another operator $O_v$ is used to model the output $\mathbf{v}_t$ in (1). Then, the hidden state $\mathbf{h}_t$ is defined to propagate the information across triplets. Taking Chains [11] as an example, $O_s$ is an adding combinator for $\mathbf{h}_{t-1}$ and $\mathbf{s}_t$, $O_r$ is an RNN cell to combine $O_s$ and $\mathbf{r}_t$, and $O_v$ directly outputs $O_r$ as $\mathbf{v}_t$.

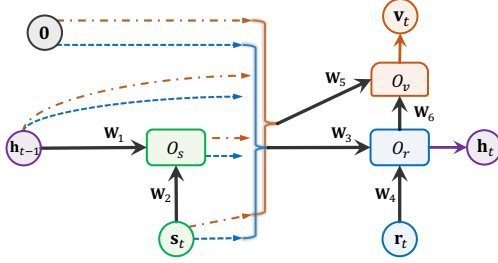

Figure 2: Search space $\mathcal{A}$ of $f$ for (1).

Table 2: The split search space of $f$ in Figure 2 into macro-level $\boldsymbol{\alpha}_1$ and micro-level $\boldsymbol{\alpha}_2$.

| macro-level | connections | $\mathbf{h}_{t-1}, O_s, \mathbf{0}, \mathbf{s}_t$ |
|---|---|---|
| $\boldsymbol{\alpha}_1 \in \mathcal{A}_1$ | combinators | $+, \odot, \otimes,$ gated |
| micro-level | activation | identity, tanh, sigmoid |
| $\boldsymbol{\alpha}_2 \in \mathcal{A}_2$ | weight matrix | $\{\mathbf{W}_i\}_{i=1}^{6}, \mathbf{I}$ |

To control the information flow, we search the connections from input vectors to the outputs, i.e. the dashed lines in Figure 2. Then, the combinators, which combine two vectors into one, are important since they determine how embedding are transformed, e.g. "$+$" in TransE [7] and "$\odot$" in DistMult [55]. As in the search space of RNN [63], we introduce activation functions *tanh* and *sigmoid* to give non-linear squashing. Each link is either a trainable weight matrix $\mathbf{W}$ or an identity matrix $\mathbf{I}$ to adjust the vectors. Detailed information is listed in Table 2.

Let the training and validation set be $\mathcal{G}_{\text{tra}}$ and $\mathcal{G}_{\text{val}}$, $\mathcal{M}$ be the measurement on $\mathcal{G}_{\text{val}}$ and $\mathcal{L}$ be the loss on $\mathcal{G}_{\text{tra}}$. To meet different requirements on $f$, we propose to search the architecture $\boldsymbol{\alpha}$ of $f$ as a special RNN. The network here is not a general RNN but one specific to the KG embedding tasks. The problem is defined to find an architecture $\boldsymbol{\alpha}$ such that validation performance is maximized, i.e.,

$$\boldsymbol{\alpha}^* = \arg\max_{\boldsymbol{\alpha} \in \mathcal{A}} \mathcal{M}\left(f(\boldsymbol{F}^*; \boldsymbol{\alpha}), \mathcal{G}_{\text{val}}\right), \quad \text{s.t.} \quad \boldsymbol{F}^* = \arg\min_{\boldsymbol{F}} \mathcal{L}\left(f(\boldsymbol{F}; \boldsymbol{\alpha}), \mathcal{G}_{\text{tra}}\right), \tag{2}$$

which is a bi-level optimization problem and is non-trivial to solve. First, the computation cost to get $\boldsymbol{F}^*$ is generally high. Second, searching for $\boldsymbol{\alpha} \in \mathcal{A}$ is a discrete optimization problem [29] and the space is large (in Appendix A.1). Thus, how to efficiently search the architectures is a big challenge.

Compared with standard RNNs, which recurrently model each input vectors, the Interstellar models the relational path with triplets as basic unit. In this way, we can determine how to model short-term information inside each triplet and what long-term information should be passed along the triplets. This makes our search space distinctive for the KG embedding problems.

## 3.2 Proposed Search Algorithm

In Section 3.1, we have introduced the search space $\mathcal{A}$, which contains considerable different architectures. Therefore, how to search efficiently in $\mathcal{A}$ is an important problem. Designing appropriate optimization algorithm for the discrete architecture parameters is a big challenge.

### 3.2.1 Problems with Existing Algorithms

As introduced in Section 2.2, we can either choose the stand-alone approach or one-shot approach to search the network architecture. In order to search efficiently, search algorithms should be designed for specific scenarios [63, 29, 1]. When the search space is complex, e.g. the macro space of CNNs and the tree-structured RNN cells [63, 36], stand-alone approach is preferred since it can provide accurate evaluation feedback while PS scheme is not reliable [5, 41]. However, the computation

cost of evaluating an architecture under the stand-alone approach is high, preventing us to efficiently search in large spaces.

One-shot search algorithms are widely used in searching micro spaces, e.g. cell structures in CNNs and simplified DAGs in RNN cells [1, 36, 29]. Searching architectures on a supernet with PS in the simplified space is relatively possible. However, since the search space in our problem is more complex and the embedding status influences the evaluation, PS is not reliable (see Appendix B.2). Therefore, one-shot algorithms is not appropriate in our problem.

### 3.2.2 Hybrid-search Algorithm

Even though the two types of algorithms have their limitations, is it possible to take advantage from both of them? Back to the development from stand-alone search in NASNet [63] to one-shot search in ENAS [36], the search space is simplified from a macro space to micro cells. We are motivated to split the space $\mathcal{A}$ into a macro part $\mathcal{A}_1$ and a micro part $\mathcal{A}_2$ in Table 2. $\boldsymbol{\alpha}_1 \in \mathcal{A}_1$ controls the connections and combinators, influencing information flow a lot; and $\boldsymbol{\alpha}_2 \in \mathcal{A}_2$ fine-tunes the architecture through activations and weight matrix. Besides, we empirically observe that PS for $\boldsymbol{\alpha}_2$ is reliable (in Appendix B.2). Then we propose a hybrid-search method that can be both fast and accurate in Algorithm 1. Specifically, $\boldsymbol{\alpha}_1$ and $\boldsymbol{\alpha}_2$ are sampled from a controller $c$ — a distribution [1, 54] or a neural network [63]. The evaluation feedback of $\boldsymbol{\alpha}_1$ and $\boldsymbol{\alpha}_2$ are obtained through the stand-along manner and one-shot manner respectively. After the searching procedure, the best architecture is sampled and we fine-tune the hyper-parameters to achieve the better performance.

---

**Algorithm 1** Proposed search recurrent architecture as the Interstellar algorithm.

**Require:** search space $\mathcal{A} \equiv \mathcal{A}_1 \cup \mathcal{A}_2$ in Figure 2, controller $c$ for sampling $\boldsymbol{\alpha} = [\boldsymbol{\alpha}_2, \boldsymbol{\alpha}_1]$.
1: **repeat**
2:      sample the *micro-level* architecture $\boldsymbol{\alpha}_2 \in \mathcal{A}_2$ by $c$;
3:      update the controller $c$ for $k_1$ steps using Algorithm 2 (in *stand-alone* manner);
4:      sample the *macro-level* architecture $\boldsymbol{\alpha}_1 \in \mathcal{A}_1$ by $c$;
5:      update the controller $c$ for $k_2$ steps using Algorithm 3 (in *one-shot* manner);
6: **until** termination
7: Fine-tune the hyper-parameters for the best architecture $\boldsymbol{\alpha}^* = [\boldsymbol{\alpha}_1^*, \boldsymbol{\alpha}_2^*]$ sampled from c.
8: **return** $\boldsymbol{\alpha}^*$ and the fine-tuned hyper-parameters.

---

In the macro-level (Algorithm 2), once we get the architecture $\boldsymbol{\alpha}$, the parameters are obtained by full model training. This ensures that the evaluation feedback for macro architectures $\boldsymbol{\alpha}_1$ is reliable. In the one-shot stage (Algorithm 3), the main difference is that, the parameters $\boldsymbol{F}$ are not initialized and different architectures are evaluated on the same set of $\boldsymbol{F}$, i.e. by PS. This improves the efficiency of evaluating micro architecture $\boldsymbol{\alpha}_2$ without full model training.

---

**Algorithm 2** Macro-level ($\boldsymbol{\alpha}_1$) update

**Require:** controller $c$, $\boldsymbol{\alpha}_2 \in \mathcal{A}_2$, parameters $\boldsymbol{F}$.
1: sample an individual $\boldsymbol{\alpha}_1$ by $c$ to get the architecture $\boldsymbol{\alpha} = [\boldsymbol{\alpha}_1, \boldsymbol{\alpha}_2]$;
2: initialize $\boldsymbol{F}$ and train to obtain $\boldsymbol{F}^*$ until converge by minimizing $L(f(\boldsymbol{F}, \boldsymbol{\alpha}), \mathcal{G}_{\text{tra}})$;
3: evaluate $\mathcal{M}(f(\boldsymbol{F}^*, \boldsymbol{\alpha}), \mathcal{G}_{\text{val}})$ to update $c$.
4: **return** the updated controller $c$.

**Algorithm 3** Micro-level ($\boldsymbol{\alpha}_2$) update

**Require:** controller $c$, $\boldsymbol{\alpha}_1 \in \mathcal{A}_1$, parameters $\boldsymbol{F}$.
1: sample an individual $\boldsymbol{\alpha}_2$ by $c$ to get the architecture $\boldsymbol{\alpha} = [\boldsymbol{\alpha}_1, \boldsymbol{\alpha}_2]$;
2: sample a mini-batch $\mathcal{B}_{\text{tra}}$ from $\mathcal{G}_{\text{tra}}$ and update $\boldsymbol{F}$ with gradient $\nabla_{\boldsymbol{F}} \mathcal{L}(f(\boldsymbol{F}, \boldsymbol{\alpha}), \mathcal{B}_{\text{tra}})$;
3: sample a mini-batch $\mathcal{B}_{\text{val}}$ from $\mathcal{G}_{\text{val}}$ and evaluate $\mathcal{M}(f(\boldsymbol{F}, \boldsymbol{\alpha}), \mathcal{B}_{\text{val}})$ to update $c$.
4: **return** the updated controller $c$.

---

The remaining problem is how to model and update the controller, i.e. step 3 in Algorithm 2 and step 3 in Algorithm 3. The evaluation metrics in KG tasks are usually ranking-based, e.g. Hit@$k$, and are non-differentiable. Instead of using the direct gradient, we turn to the derivative-free optimization methods [10], such as reinforcement learning (policy gradient in [63] and Q-learning in [4]), or Bayes optimization [6]. Inspired by the success of policy gradient in searching CNNs and RNNs [63, 36], we use policy gradient to optimize the controller $c$.

Following [1, 16, 54], we use stochastic relaxation for the architectures $\boldsymbol{\alpha}$. Specifically, the parametric probability distribution $\boldsymbol{\alpha} \sim p_{\boldsymbol{\theta}}(\boldsymbol{\alpha})$ is introduced on the search space $\boldsymbol{\alpha} \in \mathcal{A}$ and then the distribution parameter $\boldsymbol{\theta}$ is optimized to maximize the expectation of validation performance

$$\max_{\boldsymbol{\theta}} J(\boldsymbol{\theta}) = \max_{\boldsymbol{\theta}} \mathbb{E}_{\boldsymbol{\alpha} \sim p_{\boldsymbol{\theta}}(\boldsymbol{\alpha})} \left[ \mathcal{M}\left(f(\boldsymbol{F}^*; \boldsymbol{\alpha}), \mathcal{G}_{\text{val}}\right) \right]. \tag{3}$$

Then, the optimization target is transformed from (2) with the discrete $\boldsymbol{\alpha}$ into (3) with the continuous $\boldsymbol{\theta}$. In this way, $\boldsymbol{\theta}$ is updated by $\boldsymbol{\theta}_{t+1} = \boldsymbol{\theta}_t + \rho \nabla_{\boldsymbol{\theta}} J(\boldsymbol{\theta}_t)$, where $\rho$ is the step-size, and $\nabla_{\boldsymbol{\theta}} J(\boldsymbol{\theta}) = \mathbb{E} [\mathcal{M}(f(F; \boldsymbol{\alpha}), \mathcal{G}_{\text{val}}) \nabla_{\boldsymbol{\theta}} \ln (p_{\boldsymbol{\theta}_t}(\boldsymbol{\alpha}))]$ is the pseudo gradient of $\boldsymbol{\alpha}$. The key observation is that we do not need to take direct gradient w.r.t. $\mathcal{M}$, which is not available. Instead, we only need to get the validation performance measured by $\mathcal{M}$.

To further improve the efficiency, we use natural policy gradient (NPG) [34] $\tilde{\nabla}_{\boldsymbol{\theta}} J(\boldsymbol{\theta}_t)$ to replace $\nabla_{\boldsymbol{\theta}} J(\boldsymbol{\theta}_t)$, where $\tilde{\nabla}_{\boldsymbol{\theta}} J(\boldsymbol{\theta}_t) = [\boldsymbol{H}(\boldsymbol{\theta}_t)]^{-1} \nabla_{\boldsymbol{\theta}} J(\boldsymbol{\theta}_t)$ is computed by multiplying a Fisher information matrix $\boldsymbol{H}(\boldsymbol{\theta}_t)$ [2]. NPG has shown to have better convergence speed [1, 2, 34] (see Appendix A.2).

## 4 Experiments

### 4.1 Experiment Setup

Following [17, 18, 19], we sample the relational paths from biased random walks (details in Appendix A.3). We use two basic tasks in KG, i.e. entity alignment and link prediction. Same as the literature [7, 18, 52, 62], we use the "filtered" ranking metrics: mean reciprocal ranking (MRR) and Hit@$k(k = 1, 10)$. Experiments are written in Python with PyTorch framework [35] and run on a single 2080Ti GPU. Statistics of the data set we use in this paper is in Appendix A.4. Training details of each task are given in Appendix A.5. Besides, all of the searched models are shown in Appendix C due to space limitations.

### 4.2 Understanding the Search Space

Here, we illustrate the designed search space $\mathcal{A}$ in Section 3.1 using Countries [8] dataset, which contains 271 countries and regions, and 2 relations *neighbor* and *locatedin*. This dataset contains three tasks: S1 infers $neighbor \wedge locatedin \rightarrow locatedin$ or $locatedin \wedge locatedin \rightarrow locatedin$; S2 require to infer the 2 hop relations $neighbor \wedge locatedin \rightarrow locatedin$; S3 is harder and requires modeling 3 hop relations $neighbor \wedge locatedin \wedge locatedin \rightarrow locatedin$.

To understand the dependency on the length of paths for various tasks, we extract four subspaces from $\mathcal{A}$ (in Figure 2) with different connections. Specifically, (P1) represents the single hop embedding models; (P2) processes 2 successive steps; (P3) models long relational paths without intermediate entities; and (P4) includes both entities and relations along path. Then we search $f$ in P1-P4 respectively to see how well these subspaces can tackle the tasks S1-S3.

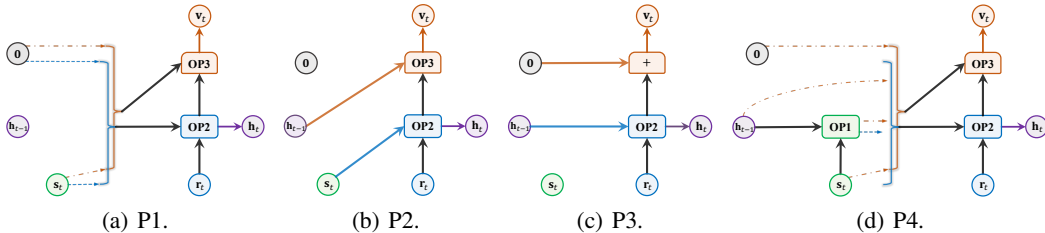

| (a) P1. | (b) P2. | (c) P3. | (d) P4. |

Figure 3: Four subspaces with different connection components in the recurrent search space.

For each task, we randomly generate 100 models for each subspace and record the model with the best *area under curve of precision recall* (AUC-PR) on validation set. This procedure is repeated 5 times to evaluate the testing performance in Table 3. We can see that there is no single subspace performing well on all tasks. For easy tasks S1 and S2, short-term information is more important. Incorporating entities along path like P4 is bad for learning 2 hop relationships. For the harder task S3, P3 and P4 outperform the others since it can model long-term information in more steps.

Table 3: Performance on Countries dataset.

|  | S1 | S2 | S3 |
|---|---|---|---|
| P1 | 0.998±0.001 | 0.997±0.002 | 0.933±0.031 |
| P2 | **1.000±0.000** | 0.999±0.001 | 0.952±0.023 |
| P3 | 0.992±0.001 | **1.000±0.000** | 0.961±0.016 |
| P4 | 0.977±0.028 | 0.984±0.010 | **0.964±0.015** |
| **Interstellar** | **1.000±0.000** | **1.000±0.000** | **0.968± 0.007** |

Besides, we evaluate the best model searched in the whole space $\mathcal{A}$ for S1-S3. As in the last line of Table 3, the model searched by Interstellar achieves good performance on the hard task S3. Interstellar prefers different candidates (see Appendix C.2) for S1-S3 over the same search space.

This verifies our analysis that it is difficult to use a unified model that can adapt to the short-term and long term information for different KG tasks.

## 4.3 Comparison with State-of-the-art KG Embedding Methods

**Entity Alignment.** The entity alignment task aims to align entities in different KGs referring the same instance. In this task, long-term information is important since we need to propagate the alignment across triplets [18, 45, 62]. We use four cross-lingual and cross-database subset from DBpedia and Wikidata generated by [18], i.e. DBP-WD, DBP-YG, EN-FR, EN-DE. For fair comparison, we follow the same path sampling scheme and the data set splits in [18].

Table 4: Performance comparison on entity alignment task. H@$k$ is short for Hit@$k$. The results of TransD [21], BootEA [45], IPTransE [62], GCN-Align [53] and RSN [18] are copied from [18].

| | models | DBP-WD | | | DBP-YG | | | EN-FR | | | EN-DE | | |
|---|---|---|---|---|---|---|---|---|---|---|---|---|---|
| | | H@1 | H@10 | MRR | H@1 | H@10 | MRR | H@1 | H@10 | MRR | H@1 | H@10 | MRR |
| triplet | TransE | 18.5 | 42.1 | 0.27 | 9.2 | 24.8 | 0.15 | 16.2 | 39.0 | 0.24 | 20.7 | 44.7 | 0.29 |
| | TransD* | 27.7 | 57.2 | 0.37 | 17.3 | 41.6 | 0.26 | 21.1 | 47.9 | 0.30 | 24.4 | 50.0 | 0.33 |
| | BootEA* | 32.3 | 63.1 | 0.42 | 31.3 | 62.5 | 0.42 | 31.3 | 62.9 | 0.42 | 44.2 | 70.1 | 0.53 |
| GCN | GCN-Align | 17.7 | 37.8 | 0.25 | 19.3 | 41.5 | 0.27 | 15.5 | 34.5 | 0.22 | 25.3 | 46.4 | 0.22 |
| | VR-GCN | 19.4 | 55.5 | 0.32 | 20.9 | 55.7 | 0.32 | 16.0 | 50.8 | 0.27 | 24.4 | 61.2 | 0.36 |
| | R-GCN | 8.6 | 31.4 | 0.16 | 13.3 | 42.4 | 0.23 | 7.3 | 31.2 | 0.15 | 18.4 | 44.8 | 0.27 |
| path | PTransE | 16.7 | 40.2 | 0.25 | 7.4 | 14.7 | 0.10 | 7.3 | 19.7 | 0.12 | 27.0 | 51.8 | 0.35 |
| | IPTransE* | 23.1 | 51.7 | 0.33 | 22.7 | 50.0 | 0.32 | 25.5 | 55.7 | 0.36 | 31.3 | 59.2 | 0.41 |
| | Chains | 32.2 | 60.0 | 0.42 | 35.3 | 64.0 | 0.45 | 31.4 | 60.1 | 0.41 | 41.3 | 68.9 | 0.51 |
| | RSN* | 38.8 | 65.7 | 0.49 | 40.0 | 67.5 | 0.50 | 34.7 | 63.1 | 0.44 | 48.7 | 72.0 | 0.57 |
| | **Interstellar** | **40.7** | **71.2** | **0.51** | **40.2** | **72.0** | **0.51** | **35.5** | **67.9** | **0.46** | **50.1** | **75.6** | **0.59** |

Table 4 compares the testing performance of the models searched by Interstellar and human-designed ones on the *Normal* version datasets [18] (the *Dense* version [18] in Appendix B.1). In general, the path-based models are better than the GCN-based and triplets-based models by modeling long-term dependencies. BootEA [45] and IPTransE [62] win over TransE [7] and PTransE [27] respectively by iteratively aligning discovered entity pairs. Chains [11] and RSN [18] outperform graph-based models and the other path-based models by explicitly processing both entities and relations along path. In comparison, Interstellar is able to search and balance the short-term and long-term information adaptively, thus gains the best performance. We plot the learning curve on DBP-WD of some triplet, graph and path-based models in Figure 4(a) to verify the effectiveness of the relational paths.

**Link Prediction.** In this task, an incomplete KG is given and the target is to predict the missing entities in unknown links [52]. We use three famous benchmark datasets, WN18-RR [13] and FB15k-237 [48], which are more realistic than their superset WN18 and FB15k [7], and YAGO3-10 [31], a much larger dataset.

Table 5: Link prediction results.

| models | WN18-RR | | | FB15k-237 | | | YAGO3-10 | | |
|---|---|---|---|---|---|---|---|---|---|
| | H@1 | H@10 | MRR | H@1 | H@10 | MRR | H@1 | H@10 | MRR |
| TransE | 12.5 | 44.5 | 0.18 | 17.3 | 37.9 | 0.24 | 10.3 | 27.9 | 0.16 |
| ComplEx | 41.4 | 49.0 | 0.44 | 22.7 | 49.5 | 0.31 | 40.5 | 62.8 | 0.48 |
| RotatE* | 43.6 | 54.2 | 0.47 | **23.3** | 50.4 | **0.32** | 40.2 | 63.1 | 0.48 |
| R-GCN | - | - | - | 15.1 | 41.7 | 0.24 | - | - | - |
| PTransE | 27.2 | 46.4 | 0.34 | 20.3 | 45.1 | 0.29 | 12.2 | 32.3 | 0.19 |
| RSN | 38.0 | 44.8 | 0.40 | 19.2 | 41.8 | 0.27 | 16.4 | 37.3 | 0.24 |
| Interstellar | **43.8** | **54.6** | **0.48** | **23.3** | **50.8** | **0.32** | **42.4** | **66.4** | **0.51** |

We search architectures with dimension 64 to save time and compare the models with dimension 256. Results of R-GCN on WN18-RR and YAGO3-10 are not available due to out-of-memory issue. As shown in Table 5, PTransE outperforms TransE by modeling compositional relations, but worse than ComplEx and RotatE since the adding operation is inferior to $\otimes$ when modeling the interaction between entities and relations [49]. RSN is worse than ComplEx/RotatE since it pays more attention to long-term information rather than the inside semantics. Interstellar outperforms the path-based methods PTransE and RSN by searching architectures that model fewer steps (see Appendix C.4). And it works comparable with the triplet-based models, i.e. ComplEx and RotatE, which are specially designed for this task. We show the learning curve of TransE, ComplEx, RSN and Interstellar on WN18-RR in Figure 4(b).

## 4.4 Comparison with Existing NAS Algorithms

In this part, we compare the proposed Hybrid-search algorithm in Interstellar with the other NAS methods. First, we compare the proposed algorithm with stand-alone NAS methods. Once an architecture is sampled, the parameters are initialized and trained into converge to give reliable feedback. Random search *(Random)*, Reinforcement learning *(Reinforce)* [63] and Bayes optimization *(Bayes)* [6] are chosen as the baseline algorithms. As shown in Figure 5 (entity alignment on DBP-WD and link prediction on WN18-RR), the Hybrid algorithm in Interstellar is more efficient since it takes advantage of the one-shot approach to search the micro architectures in $\mathcal{A}_2$.

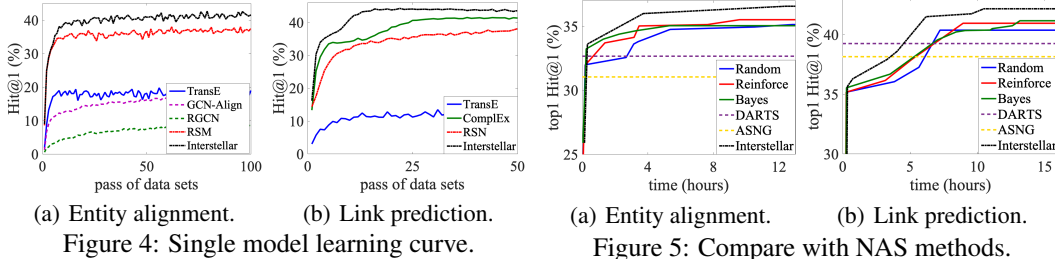

|  |  |
|---|---|
| (a) Entity alignment. | (b) Link prediction. |
| Figure 4: Single model learning curve. | |

|  |  |
|---|---|
| (a) Entity alignment. | (b) Link prediction. |
| Figure 5: Compare with NAS methods. | |

Then, we compare with one-shot NAS methods with PS on the entire space. DARTS [29] and ASNG [1] are chosen as the baseline models. For DARTS, the gradient is obtained from loss on training set since validation metric is not differentiable. We show the performance of the best architecture found by the two one-shot algorithms. As shown in the dashed lines, the architectures found by one-shot approach are much worse than that by the stand-alone approaches. The reason is that PS is not reliable in our complex recurrent space (more experiments in Appendix B.2). In comparison, Interstellar is able to search reliably and efficiently by taking advantage of both the stand-alone approach and the one-shot approach.

## 4.5 Searching Time Analysis

We show the clock time of Interstellar on entity alignment and link prediction tasks in Table 6. Since the datasets used in entity alignment task have similar scales (see Appendix A.4), we show them in the same column. For each task/dataset, we show the computation cost of the macro-level and micro-level in Interstellar for 50 iterations between step 2-5 in Algorithm 1 (20 for YAGO3-10 dataset); and the fine-tuning procedure after searching for 50 groups of hyper-parameters, i.e. learning rate, decay rate, dropout rate, L2 penalty and batch-size (details in Appendix A.5). As shown, the entity alignment tasks take about 15-25 hours, while link prediction tasks need about one or more days due to the larger data size. The cost of the search process is at the same scale with that of the fine-tuning time, which shows the search process is not expensive.

Table 6: Comparison of searching and fine-tuning time (in hours) in Algorithm 1.

| procedure | | entity alignment | | link prediction | | |
|---|---|---|---|---|---|---|
| | | Normal | Dense | WN18-RR | FB15k-237 | YAGO3-10 |
| search | macro-level (line 2-3) | 9.9±1.5 | 14.9±0.3 | 11.7±1.9 | 23.2±3.4 | 91.6±8.7 |
| | micro-level (line 4-5) | 4.2±0.2 | 7.5±0.6 | 6.3± 0.9 | 5.6±0.4 | 10.4±1.3 |
| fine-tune (line 7) | | 11.6±1.6 | 16.2±2.1 | 44.3±2.3 | 67.6±4.5 | > 200 |

## 5 Conclusion

In this paper, we propose a new NAS method, Interstellar, to search RNN for learning from the relational paths, which contain short-term and long-term information in KGs. By designing a specific search space based on the important properties in relational path, Interstellar can adaptively search promising architectures for different KG tasks. Furthermore, we propose a hybrid-search algorithm that is more efficient compared with the other state-of-art NAS algorithms. The experimental results verifies the effectiveness and efficiency of Interstellar on various KG embedding benchmarks. In future work, we can combine Interstellar with AutoSF [61] to give further improvement on the embedding learning problems. Taking advantage of data similarity to improve the search efficiency on new datasets is another extension direction.

# Broader impact

Most of the attention on KG embedding learning has been focused on the triplet-based models. In this work, we emphasis the benefits and importance of using relational paths to learn from KGs. And we propose the path-interstellar as a recurrent neural architecture search problem. This is the first work applying neural architecture search (NAS) methods on KG tasks.

In order to search efficiently, we propose a novel hybrid-search algorithm. This algorithm addresses the limitations of stand-alone and one-shot search methods. More importantly, the hybrid-search algorithm is not specific to the problem here. It is also possible to be applied to the other domains with more complex search space [63, 47, 42].

One limitation of this work is that, the Interstellar is currently limited on the KG embedding tasks. Extending to reasoning tasks like DRUM [38] is an interesting direction.

# Acknowledgment

This work is partially supported by National Key Research and Development Program of China Grant No. 2018AAA0101100, the Hong Kong RGC GRF Project 16202218, CRF Project C6030-18G, C1031-18G, C5026-18G, AOE Project AoE/E-603/18, China NSFC No. 61729201, Guangdong Basic and Applied Basic Research Foundation 2019B151530001, Hong Kong ITC ITF grants ITS/044/18FX and ITS/470/18FX. Lei Chen is partially supported by Microsoft Research Asia Collaborative Research Grant, Didi-HKUST joint research lab project, and Wechat and Webank Research Grants.

## Footnotes

[1]Code is available at `https://github.com/AutoML-4Paradigm/Interstellar`, and correspondence is to Q. Yao.

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
