[Supplementary Material]

# A Supplementary Details

## A.1 Details of the search space

**Operators**  Given two input vectors $\mathbf{a}$ and $\mathbf{b}$ with $d$-dimensions and $d$ is an even number. Two basic combinators are (a). adding ($+$): $\mathbf{o}_i = \mathbf{a}_i + \mathbf{b}_i$; and (b).multiplying ($\odot$): $\mathbf{o}_i = \mathbf{a}_i \cdot \mathbf{b}_i$. In order to cover ComplEx [49] in the search space, we use Hermitian product ($\otimes$) as in [49]

$$\mathbf{o}_i = \begin{cases} \mathbf{a}_i \cdot \mathbf{b}_i - \mathbf{a}_{i+d/2} \cdot \mathbf{b}_{i+d/2} & \text{if } i < d/2 \\ \mathbf{a}_{i-d/2} \cdot \mathbf{b}_i - \mathbf{a}_i \cdot \mathbf{b}_{i-d/2} & \text{otherwise} \end{cases}$$

As for the gated unit, we define it as $\mathbf{o}_i = \mathbf{g}_i \cdot \mathbf{a}_i + (1 - \mathbf{g}_i) \cdot \mathbf{b}_i$, where the gate function $\mathbf{g} = \text{sigmoid}(\mathbf{W}_a\mathbf{a} + \mathbf{W}_b\mathbf{b})$ with trainable parameters $\mathbf{W}_a, \mathbf{W}_b \in \mathbb{R}^{d \times d}$, to imitate the gated recurrent function.

**Space size**  Based on the details in Table 2, the inputs of $O_r$ and $O_v$ are chosen from $\mathbf{h}_{t-1}, O_s, \mathbf{0}, \mathbf{s}_t$. Then the three operators $O_s, O_r, O_v$ select one combinator from $+, \odot, \otimes$ and gated. Thus, we have $4^2 \times 4^3 = 1024$ architectures in the macro space $\mathcal{A}_1$. For the micro space, we only use activation functions after $O_s, O_r$ since we empirically observe that activation function on $O_v$ is bad for the embedding spaces. In this way, the size of micro space $\mathcal{A}_2$ is $3^2 \times 2^6 = 576$. Totally, there should be $6 \times 10^5$ candidates with the combination of macro-level space and micro-level space.

## A.2 Details of the search algorithm

**Compare with existing NAS problem.**  In Section 3.2, we have already talked about the problems of existing search algorithms and propose a hybrid-search algorithm that can search fast and accurate. Here, we give an overview of the different NAS methods in Table 7.

Table 7: Comparison of state-of-the-art NAS methods for general RNN with the proposed Interstellar.

|  | Existing NAS Algorithms | | Interstellar |
|---|---|---|---|
|  | Stand-alone approach | One-shot approach |  |
| space | complex structures | micro cells | KG-specific |
| algorithm | reinforcement learning [63], Bayes optimization [6, 22] | direct gradient descent [29], stochastic relaxation [1, 54] | natural policy gradient |
| evaluation | full model training | parameter-sharing | hybrid |

**Implementation of $\mathcal{M}(f(\boldsymbol{F}^*, \boldsymbol{\alpha}), \mathcal{G}_{\textbf{val}})$.**  For both tasks, the training samples are composed of relational paths. But the validation data format is different. For the entity alignment task, the validation data is a set of entity pairs, i.e. $\mathcal{S}_{val} = \{(e_1, e_2)|e_1 \in \mathcal{E}_1, e_2 \in \mathcal{E}_2\}$. The performance is measured by the cosine similarity of the embeddings $\mathbf{e}_1$ and $\mathbf{e}_2$. Thus, the architecture $\boldsymbol{\alpha}$ is not available in $\mathcal{M}(f(\boldsymbol{F}^*, \boldsymbol{\alpha}), \mathcal{G}_{\text{val}})$. Instead, we use the negative loss function on a training mini-batch $-\mathcal{L}(f(\boldsymbol{F}^*, \boldsymbol{\alpha}), \mathcal{B}_{\text{tra}})$ as an alternative of $\mathcal{M}(f(\boldsymbol{F}^*, \boldsymbol{\alpha}), \mathcal{G}_{\text{val}})$. For the link prediction task, the validation data is set of single triplets, i.e. 1-step paths. Different from the entity alignment task, the path is available for validation measurement here. Therefore, once we sample a mini-batch $\mathcal{B}_{\text{val}}$ from $\mathcal{G}_{\text{val}}$, we can either use the loss function $-\mathcal{L}(f(\boldsymbol{F}^*, \boldsymbol{\alpha}), \mathcal{B}_{\text{val}})$ or the direct evaluation metric (MRR or Hit@$k$) $\mathcal{M}(f(\boldsymbol{F}^*, \boldsymbol{\alpha}), \mathcal{B}_{\text{val}})$ on the mini-batch $\mathcal{B}_{\text{val}}$.

**Implementation of natural policy gradient.**  Recall that the optimization problem of architectures is changed from optimizing $\boldsymbol{\alpha}$ by $\boldsymbol{\alpha}^* = \arg\max_{\boldsymbol{\alpha} \in \mathcal{A}} \mathcal{M}(f(\boldsymbol{F}^*; \boldsymbol{\alpha}), \mathcal{G}_{\text{val}})$ to optimizing the distribution parameter $\boldsymbol{\theta}$ by $\max_{\boldsymbol{\theta}} J(\boldsymbol{\theta}) = \max_{\boldsymbol{\theta}} \mathbb{E}_{\boldsymbol{\alpha} \sim p_{\boldsymbol{\theta}}(\boldsymbol{\alpha})} [\mathcal{M}(f(\boldsymbol{F}^*; \boldsymbol{\alpha}), \mathcal{G}_{\text{val}})]$. As discussed in Section 3.2.2, we use the natural policy gradient to solve $\boldsymbol{\theta}^* = \arg\max_{\boldsymbol{\theta}} J(\theta)$. To begin with, we need to define the distribution $p_{\boldsymbol{\theta}}$. Same as [1, 33], we refer to the exponential family $h(\boldsymbol{\alpha}) \cdot \exp(\eta(\boldsymbol{\theta})^\top T(\boldsymbol{\alpha}) - A(\boldsymbol{\theta}))$, where $h(\boldsymbol{\alpha}), T(\boldsymbol{\alpha})$ and $A(\boldsymbol{\theta})$ are known functions depending on the target distribution. The benefit of using exponential family is that the inverse Fisher information matrix $\boldsymbol{H}^{-1}(\boldsymbol{\theta})$ is easily obtained. For simplicity, we set $h(\boldsymbol{\alpha}) = 1$ and choose the expectation

parameters $\boldsymbol{\theta} = \mathbb{E}_{p_{\boldsymbol{\theta}}}[T(\boldsymbol{\alpha})]$ as in [18]. Then the gradient reduces to $\nabla_{\boldsymbol{\theta}} \ln(p_{\boldsymbol{\theta}}(\boldsymbol{\alpha})) = T(\boldsymbol{\alpha}) - \boldsymbol{\theta}$, and the inverse Fisher information matrix $\boldsymbol{H}^{-1}(\boldsymbol{\theta}) = \mathbb{E}_{p_{\boldsymbol{\theta}}}[(T(\boldsymbol{\alpha}) - \theta)(T(\boldsymbol{\alpha}) - \theta)^{\top}]$ is computed with the finite approximation. The iteration is,

$$\boldsymbol{\theta}_{t+1} = \boldsymbol{\theta}_t + \rho \frac{1}{m} \sum_{i=1}^{m} \left[ \left( T(\boldsymbol{\alpha}^{(i)}) - \boldsymbol{\theta}_t \right) \left( T(\boldsymbol{\alpha}^{(i)}) - \boldsymbol{\theta}_t \right)^{\top} \cdot \mathcal{M}\left( f(\boldsymbol{F}^{(i)^*}; \boldsymbol{\alpha}^{(i)}), \mathcal{G}_{\text{val}} \right) \right],$$

where $\boldsymbol{\alpha}^{(i)}$'s are sampled from $p_{\boldsymbol{\theta}_t}$. In this paper, we set the finite sample $m$ as 2 to save time. Since the architecture parameters are categorical, we model $p_{\boldsymbol{\theta}}$ as categorical distributions. For a component with $K$ categories, we use a $K$ dimensional vector $\boldsymbol{\theta}$ to model the distribution $p_{\boldsymbol{\theta}}$. The probability of sampling the $i$-th category is $\boldsymbol{\theta}_i$ and $\theta_K = 1 - \sum_{j=1}^{K-1} \boldsymbol{\theta}_j$. Note that, the NG is only used to update the first $K-1$ dimension of $\boldsymbol{\theta}$, and $T(\boldsymbol{\alpha}_i)$ is a one-hot vector (without dimension $K$) of the category $\boldsymbol{\alpha}_i$. For more details, please refer to Section 2.4 in [18] and the published code.

### A.3 Training details

Following [17, 18], we sample paths from biased random walks. Take Figure 6 as an example and the path walked from $e_0$ to $e_1$ just now. In conventional random walk, all the neighbors of $e_1$ have the same probability to be sampled as the next step. In biased random walk, we sample the neighbor that can go deeper or go to another KG with larger probability. For single KG, we use one parameter $\alpha \in (0.5, 1)$ to give the depth bias. Specifically, the neighbors that are two steps away from the previous entity $e_0$, like $e_2$ and $e_4$ in Figure 6, have the bias of $\alpha$. And the bias of others, i.e. $e_0$, $e_3$, are $1 - \alpha$. Since $\alpha > 0.5$, the path is more likely to go deeper. Similarly, we have another parameter $\beta \in (0.5, 1)$ for two KGs to give cross-KG bias. Assume $e_0$, $e_1$, and $e_3$ belongs to the same KG $\mathcal{G}_1$ and $e_2$, $e_4$ are parts of $\mathcal{G}_2$. Then, the next step is encouraged to jump to entities in another KG, namely to the entities $e_2$ and $e_4$ in Figure 6. Since the aligned pairs are usually rare in the training set, encouraging cross-KG walk can learn more about the aligned entities in the two KGs.

Figure 6: An example of the biased random walks.

Since the KGs are usually large, we sample two paths for each triplet in $\mathcal{S}_{tra}$. The length of paths is 7 for entity alignment task on two KGs and 3 for link prediction task on single KG. The paths are fixed for each dataset to keep a fair comparison among different models. The parameters used for sampling path are summarized in Table 8.

Table 8: Parameter used for sampling path.

| parameters | $\alpha$ | $\beta$ | length |
|---|---|---|---|
| entity alignment | 0.9 | 0.9 | 7 |
| link prediction | 0.7 | – | 3 |

Once the paths are sampled, we start training based on the relational path. The loss for a path with length $L$, i.e. containing $L$ triplets, is given in given as

$$\mathcal{L}_{tra} = \sum_{t=1}^{L} \left\{ -\mathbf{v}_t \cdot \mathbf{o}_t + \log \left( \sum_{o_i \in \mathcal{E}} \exp\left(\mathbf{v}_t \cdot \mathbf{o}_i\right) \right) \right\} \tag{4}$$

In the recurrent step $t$, we focus on one triplet $(s_t, r_t, o_t)$. The subject entity embedding $\mathbf{s}_t$ and relation embedding $\mathbf{r}_t$ are processed along with the proceeding information $\mathbf{h}_{t-1}$ to get the output $\mathbf{v}_t$. The output $\mathbf{v}_t$ is encouraged to approach the object entity embedding $\mathbf{o}_t$. Thus, the objective in (4) can be regarded as a multi-class log-loss [25] where the object $o_t$ is the true label.

Besides, the training of the parameters $\boldsymbol{F}$ is based on mini-batch gradient descent. Adam [23] is used as the optimization method for updating model parameters.

## A.4 Data statistics

The datasets used in entity alignment task are the customized version by [18]. For the normal version data, nodes are sampled to approximate the degree distribution of original KGs. This makes the datasets in this version more realistic. For the dense version data, the entities with low degrees are randomly removed in the original KGs. This makes the data more similar to those used by existing methods [45, 53]. We give the statistics in Table 9.

Table 9: Statistics of the datasets we used for entity alignment. Each single KG contains 15,000 entities. There are 4,500 aligned entity pairs in the training sets, and 11,500 pairs for evaluation. The first 10% pairs among the 11,500 are used for validation and the left for testing. We use "#" short for "number of".

| version | data source | DBP-WD | | DBP-YG | | EN-FR | | EN-DE | |
|---|---|---|---|---|---|---|---|---|---|
| | | DBpedia | Wikidata | DBpedia | YAGO3 | English | French | English | German |
| Normal | # relations | 253 | 144 | 219 | 30 | 211 | 177 | 225 | 118 |
| | # triplets | 38,421 | 40,159 | 33,571 | 34,660 | 36,508 | 33,532 | 38,281 | 37,069 |
| Dense | # relations | 220 | 135 | 206 | 30 | 217 | 174 | 207 | 117 |
| | # triplets | 68,598 | 75,465 | 71,257 | 97,131 | 71,929 | 66,760 | 56,983 | 59,848 |

The datasets used for the link prediction task are used in many baseline models [13, 44, 18]. The WN18-RR and FB15k-237 datasets are created so that the link leakage problem [13, 49] can be solved. Thus, they are more realistic than their super-set WN18 and FB15k [7].

Table 10: Statistics of the datasets used for link prediction.

| Dataset | #entity | #relations | #train | #valid | #test |
|---|---|---|---|---|---|
| WN18-RR | 40,943 | 11 | 86,835 | 3,034 | 3,134 |
| FB15k-237 | 14,541 | 237 | 272,115 | 17,535 | 20,466 |
| YAGO3-10 | 123,188 | 37 | 1,079,040 | 5,000 | 5,000 |

## A.5 Hyper-parameters and Training Details

During searching, the meta hyper-parameters $k_1$ is one step and $k_2$ is one epoch based on the efficiency consideration.

In order to find the hyper-parameters during search procedure, we use RSN [18] as a standard baseline to tune the hyper-parameters. We use the *HyperOpt* package with tree parsen estimator [6] to search the learning rate $\eta$, L2 penalty $\lambda$, decay rate $u$, batch size $m$, as well as a dropout rate $p$. The decay rate is applied to the learning rate each epoch and the dropout rate is applied to the input embeddings. The tuning ranges are given in Table 11.

Table 11: Searching range of hyper-parameters

| hyper-param | ranges |
|---|---|
| $\eta$ | $[10^{-5}, 10^{-3}]$ |
| $\lambda$ | $[10^{-5}, 10^{-2}]$ |
| $u$ | $[0.98, 1]$ |
| $m$ | $\{128, 256, 512, 1024, 2048\}$ |
| $p$ | $[0, 0.6]$ |

The embedding dimension for entity alignment task is 256, and for link prediction is 64 during searching. After the hyper-parameter tuning, the proposed Interstellar starts searching with this hyper-parameter setting. For all the tasks and datasets, we search and evaluate 100 architectures. Among them, we select the best architecture indicated by the Hit@1 performance on validation set, i.e. the architecture with top 1 performance in the curve of Interstellar in Figure 5. After searching, the embedding size on link prediction is increased to be 256 and we fine-tune the other hyper-parameters in Table 11 again for the searched architectures. Finally, the performance on testing data is reported in Table 4. The time cost of searching and fine-tuning is given in Table 6 in Section 4.5.

# B Supplementary Experiments

## B.1 Entity alignment on the Dense version

Table 12 compares the testing performance of the models searched by Interstellar and human-designed ones on the *Dense* version datasets [18]. In this version, there is no much gap between the triplet-based and GCN-based models since more data can be used for learning the triplet-based ones. In comparison, the path-based models are better by traversing across two KGs. Among them, Interstellar performs the best on DBP-WD, EN-FR and EN-DE and is comparable with RSN on DBP-YG dataset. Besides, the searched architectures are different between the Normal version and Dense version, as will be illustrated Appendix C.3.

Table 12: Performance comparison on entity alignment tasks on dense dataset. H@$k$ is short for Hit@$k$. The results of TransD [21], BootEA [45], IPTransE [62], GCN-Align [53] and RSN [18] are copied from [18].

| models | | DBP-WD | | | DBP-YG | | | EN-FR | | | EN-DE | | |
|---|---|---|---|---|---|---|---|---|---|---|---|---|---|
| | | H@1 | H@10 | MRR | H@1 | H@10 | MRR | H@1 | H@10 | MRR | H@1 | H@10 | MRR |
| | TransE | 54.9 | 75.5 | 0.63 | 33.8 | 68.5 | 0.46 | 22.3 | 39.8 | 0.28 | 30.2 | 60.2 | 0.40 |
| triplet | TransD* | 60.5 | 86.3 | 0.69 | 62.1 | 85.2 | 0.70 | 54.9 | 86.0 | 0.66 | 57.9 | 81.6 | 0.66 |
| | BootEA* | 67.8 | 91.2 | 0.76 | 68.2 | 89.8 | 0.76 | 64.8 | 91.9 | 0.74 | 66.5 | 87.1 | 0.73 |
| | GCN-Align | 43.1 | 71.3 | 0.53 | 31.3 | 57.5 | 0.40 | 37.3 | 70.9 | 0.49 | 32.1 | 55.2 | 0.40 |
| GCN | VR-GCN | 39.1 | 73.1 | 0.51 | 28.5 | 59.3 | 0.39 | 35.3 | 69.6 | 0.47 | 31.5 | 61.1 | 0.41 |
| | R-GCN | 24.1 | 61.4 | 0.36 | 31.6 | 57.3 | 0.40 | 25.7 | 59.0 | 0.37 | 28.6 | 52.4 | 0.37 |
| | PTransE | 61.6 | 81.7 | 0.69 | 62.9 | 83.2 | 0.70 | 38.4 | 80.2 | 0.52 | 37.7 | 65.8 | 0.48 |
| | IPTransE* | 43.5 | 74.5 | 0.54 | 23.6 | 51.3 | 0.33 | 42.9 | 78.3 | 0.55 | 34.0 | 63.2 | 0.44 |
| path | Chains | 64.9 | 88.3 | 0.73 | 71.3 | 91.8 | 0.79 | 70.6 | 92.6 | 0.78 | 70.7 | 86.4 | 0.76 |
| | RSN* | 76.3 | 92.4 | 0.83 | **82.6** | 95.8 | **0.87** | 75.6 | 92.5 | 0.82 | 73.9 | 89.0 | 0.79 |
| | **Interstellar** | **77.9** | **94.1** | **0.84** | 81.8 | **96.2** | **0.87** | **77.3** | **94.7** | **0.84** | **75.3** | **90.3** | **0.81** |

## B.2 Problem under parameter-sharing

In this part, we empirically show the problem under parameter-sharing in the full search space $\mathcal{A}$ and how the micro-level space $\mathcal{A}_2$ solves this problem. Following [5, 59], we train the supernet in Figure 2 with parameters-sharing and 1) randomly sample 100 architectures from $\mathcal{A}$ (one-shot); 2) re-train these sampled architectures from scratch (standard-alone). If the top-performed architectures selected by one-shot approach also perform well by the stand-alone approach, then parameter-sharing works here and vice the visa. Based on the discussion in Appendix A.2, we use the loss on validation mini-batch as the measurement in the one-shot approach for entity alignment task. And we use the metric, i.e. Hit@1, on validation mini-batch as the measurement in the stand-alone approach for link prediction task. Figure 7 shows such a comparison in the full space $\mathcal{A}$. As shown in the upper-right corner of each figure, the top-performed architectures in one-shot approach do not necessarily perform the best in stand-alone approach. Therefore, searching architectures by one-shot approach in the full search space does not work since parameter-sharing does not work here.

(a) Entity alignment (DBP-WD).
(b) Link prediction (WN18-RR).

Figure 7: Comparison of models trained with one-shot model and stand-alone model on the full search space.

In order to take advantage of the efficiency in one-shot approach, we split the search space into a macro-level space $\mathcal{A}_1$ and micro-level space $\mathcal{A}_2$. To show how parameter-sharing works in the micro-level space, we use the best macro-level architecture $\alpha_1^*$ during macro-level searching, and sample 100 architectures $\alpha_2 \in \mathcal{A}_2$ to get the architecture $\alpha = [\alpha_1^*, \alpha_2]$. As shown in Figure 8, the top performed architectures in one-shot approach also perform the best in stand-alone approach. This observation verifies that 1) parameter-sharing is more likely to work in a simpler search space as discussed in [5, 36, 59]; 2) the split of the macro and micro level space in our problem is reasonable; 3) searching architectures by the Hybrid-search algorithm takes the advantage of the stand-alone approach and the one-shot approach.

(a) Entity alignment (DBP-WD).
(b) Link prediction (WN18-RR).

Figure 8: Comparison of models trained with one-shot model and stand-alone model on the micro-level search space.

## B.3  Influence of path length

In this part, we show the performance of the best architecture on each datasets with varied maximum length of the relational path in Figure 9. For entity alignment task, we vary the length from 1 to 10 for DBP-WD datasets (normal version); for link prediction task, we vary the length from 1 to 7 for WN18-RR datasets. We plot them in the same figure to make better comparison. As shown, when path length is 1, the model performs very bad. However, it has little influence on WN18-RR. This verifies that long-term information is important for the entity alignment task while short-term information is important for link prediction task. For the other length, the performance is quite stable.

Figure 9: The influence of path length.

## C  Searched Models

### C.1  Architectures in literature

(a) TransE.  (b) PTransE.  (c) Chains.  (d) RSN.

Figure 10: Graphical illustration of TransE, PTransE, Chains, RSN. TransE is a triplet-based model thus it is non-recurrent. PTransE recurrently processes the relational path without the intermediate entities. Chains simultaneously combines the entities and relations along the path, and RSN designs a customized variant of RNN. Each edge is an identity mapping, except an explicit mark of the weight matrix $\mathbf{W}_i$. Activation functions are shown in the upper right corner of the operators.

### C.2  Countries

We give the graphical illustration of architectures searched in Countries dataset in Figure 11. The search procedure is conducted in the whole search space rather than the four patterns P1-P4 in Figure 3. We can see that in Figure 11, (a) belongs to P1, (b) belongs to P2 and (c) belongs to P4. These results further verify that Interstellar can adaptively search architectures for specific KG tasks and datasets.

(a) S1  (b) S2  (c) S3

Figure 11: Graphical representation of the searched $f$ on countries dataset.

## C.3 Entity Alignment

The searched models by Interstellar, which consistently perform the best on each dataset, are graphically illustrated in Figure 12. As shown, all of the searched recurrent networks processing recurrent information in $\mathbf{h}_{t-1}$, subject entity embedding $\mathbf{s}_t$ and relation embedding $\mathbf{r}_t$ together. They have different connections, different composition operators and different activation functions, even though the searching starts with the same random seed *1234*.

More interestingly, the searched architectures in DBP-WD and DBP-YG can model multiple triplets simultaneously. While the architectures in EN-FR and EN-DE models two adjacent triplets each time. The reason is that, in the first two datasets, the KGs are from different sources, thus the distribution is various [18]. In comparison, EN-FR and EN-DE are two cross-lingual datasets, which should have more similar distributions in each KG. Therefore, we need to model the longer term information on DBP-WD and DBP-YG. In comparison, modeling the shorter term information like two triplets together is better for EN-FR and EN-DE. In this way, the model can focus more on learning short-term semantic in a range of two triplets.

(a) DBP-WD    (b) DBP-YG    (c) EN-FR    (d) EN-DE

Figure 12: Graphical representation of the searched recurrent network $f$ on each datasets in entity alignment task (Normal version).

(a) DBP-WD    (b) DBP-YG    (c) EN-FR    (d) EN-DE

Figure 13: Graphical representation of the searched recurrent network $f$ on each datasets in entity alignment task (Dense version).

## C.4 Link Prediction

The best architectures searched in link prediction tasks are given in Figure 14. In order to illustrate the models we searched, we make a statistics of the distance, i.e. the shortest path distance when regarding the KG as a simple undirected graph, between two entities in the validation and testing set in Table 13. As shown, most of the triplets have distance less than 3. Besides, as indicated by the performance on the two datasets, we infer that triplets far away from 3-hop are very challenging to model. At least in this task, triplets less or equal than 3 hops are the main focus for different models. This also explains why RSN, which processes long relational path, does not perform well in the link prediction task. The searched models in Figure 14 do not directly consider long-term structures. Instead, the architecture on WN18-RR models one triplet, and the architecture on FB15k-237 focuses on modeling two consecutive triplets. These are consistent with the statistics in Table 13, where WN18-RR has more one-hop and FB15k-237 has more two-hop triplets.

(a) WN18-RR           (b) FB15k-237

Figure 14: Graphical representation of the searched $f$ on each datasets in link prediction task.

Table 13: Percentage of the $n$-hop triplets in validation and testing datasets.

| Datasets | | Hops | | | |
|---|---|---|---|---|---|
| | | $\leq 1$ | 2 | 3 | $\geq 4$ |
| WN18-RR | validation | 35.5% | 8.8% | 22.2% | 33.5% |
| | testing | 35.0% | 9.3% | 21.4% | 34.3% |
| FB15k-237 | validation | 0% | 73.2% | 26.1% | 0.7% |
| | testing | 0% | 73.4% | 26.8% | 0.8% |