[Reviews · NeurIPS 2020]

Review 1

Summary and Contributions: The paper proposed SRAP, Search Recurrent Architectures as the Path-interstellar, which defines the path interstellar as a recurrent neural architecture search problem for the short-term and long-term information to learn from the relational path.

Strengths: 1. The paper analyzes the difficulty and importance of using the relational path to learn the short-term and long-term information in KGs. 2. The paper proposes a domain-specific search space for SRAP. 3. The paper designs a Hybrid-search algorithm to search efficiently.

Weaknesses: 1. Section 3.2.2 "Hybird-search Algorithm". Should this be "Hybrid-search Algorithm"? 2. As the proposed method needs to search through paths for long-term information, it's like to find conflict facts or relations. How to deal with the conflict is not mentioned in the paper. 3. It would be interesting to include 2 extra methods, SRAP with only macro and SRAP with only micro, to illustrate the effectiveness of macro/micro split.

Correctness: Yes

Clarity: Yes

Relation to Prior Work: Some relevant work was not compared.

Reproducibility: Yes

Additional Feedback:


Review 2

Summary and Contributions: This paper proposes to search the "Path interstellar" (a recurrent architecture) for learning representations in knowledge graphs. It solves the problem of learning short/long-term information on a relational path, which is extremely important for representing KGs. The path interstellar can adapt well to different tasks as indicated by the thorough experiments based on the well-defined search space and the novel hybrid search algorithm.

Strengths: - Clear motivation. The motivation for defining “path interstellar” is strong and clearly stated. By comparing the learning ability of triplet-based, path-based, and GCN-based methods, the path interstellar (Definition 1) is proposed as the basic model to learn from KGs. This motivation has also been verified by a case study on synthetic data (experiments in section 4.2). - Domain-specific and well-defined search space. The authors propose a novel recurrent search space specific for the path learning problem. The searched components are either motivated by the models in the literature (combinators, activations) or by the learning problem (connections). Then the short/long-term information can be controlled by search in the space. - Novel search algorithm. The hybrid search algorithm is a novel and attractive solution to solve the efficiency problem of stand-alone search and robustness of the one-shot search. The analysis of parameter-sharing and the hybrid solution may have a border impact on the broad neural architecture search area. - Extensive experiments. Apart from the case study in section 4.2, the authors show the superior of the searched models of SRAP on entity alignment and link prediction tasks in section 4.3. The efficiency of the hybrid-search algorithm is shown in section 4.4, as well as the time analysis in section 4.5. Additional experiments to understand the problem of parameter-sharing are also clearly given in the appendix B.2.

Weaknesses: - Even though the efficiency of the search algorithm is improved. The search cost still takes tens of hours. It seems that the search algorithm is conducted independently on each data set. However, the data statistics indicate some similarities among them, e.g., DBP-WD, DBP-YG. Considering the similarity should be helpful for this problem. - In the setup, the paths are sampled from the graph and then learned by the searched models. It is not sure whether the search problem will be influenced when the paths change.

Correctness: Yes

Clarity: Yes

Relation to Prior Work: Yes. - Theory: The difference with previous knowledge graph embedding models are shown in Table 1. And the difference with the previous NAS problem is discussed in section 3. - Practice: The empirical results are compared with both embedding models in section 4.3 and NAS algorithms in section 4.4.

Reproducibility: Yes

Additional Feedback: Overall, this paper is clearly written, well-motivated, and strongly novel work to learn from knowledge graphs. Here are some points and questions that could make further improvements. 1. It is well-known that the natural policy gradient is costly. The authors can add some discussion on the efficiency problem of conducting the natural gradient algorithm. 2. Path ranking algorithm in “Random Walk Inference and Learning in A Large Scale Knowledge Base” is a well-known method to learn paths in knowledge graphs. It’s better to include this paper as a reference. 3. In the experiment setup, why not sample the path and train the model simultaneously? 4. If the graph evolves, e.g., adding edges, entities, or relations, can this algorithm adapt to the new one? ----------------------------- I have read the author's rebuttal, and decide not to change my score.


Review 3

Summary and Contributions: The paper presents a neural architecture search (NAS) model for path-based KG embedding. The authors also propose a hybird-search algorithm to search optimal connection patterns from input path embeddings to the output representations. The novelty of this model lies in that it can adaptively use different network architectures to model short-term or long-term dependency in KG paths. The experiments demonstrate the effectiveness of the proposed KG embedding model and search algorithm.

Strengths: To me, the proposed neural architecture search (NAS) model for path-based KG embedding is interesting and novel. It is also relevant to NeurIPS. The experiments are extensive, including two popular KG embedding tasks on the corresponding benchmark datasets. The results show that the proposed model outperforms the SOTA path-based KG embedding models on both tasks.

Weaknesses: It seems that the proposed model does not do well in terms of efficiency. It costs a lot of time for the entity alignment and link prediction tasks on the small benchmark datasets. How does it scale to large datasets? I think this is the biggest weakness of the proposed model.

Correctness: Overall, the claims and the proposed method are reasonable. The experiments are conducted on two popular KG embedding tasks (link prediction and entity alignment) using benchmark datasets. One concern I have regarding the experiments is that some new models for link prediction or entity alignment, e.g., CompGCN (although it is not path-based model), are proposed in the recent years and it would be better to add them for comparison to make the paper more convincing.

Clarity: Overall, this paper is well written. But some sentences are somewhat difficult for me to understand. For example, lines 31-32 say "based on the relational paths, the triplet-based models work on every triplet (s_i, r_i, o_i) individually". What does it mean? It would be better to clarify such sentences.

Relation to Prior Work: Yes. The authors claim that the paper is the first work applying neural architecture search methods on KG embedding tasks, and the related work section also provides a detailed introduction about both sides. Table 1 also gives a clear comparison on how the proposed method differs from existing KG embedding models. Good Job! By the way, the citation numbers of R-GCN and GCN-Align in Table 1 are missing. Besides, I also know a related work AutoSF, which should be added for a discussion or comparison. Yongqi Zhang, Quanming Yao, Wenyuan Dai, Lei Chen: AutoSF: Searching Scoring Functions for Knowledge Graph Embedding. ICDE 2020: 433-444

Reproducibility: Yes

Additional Feedback: Please see the detailed comments. === after rebuttal === I have read the authors' response. Thanks for addressing my concerns.


Review 4

Summary and Contributions: This research applied neural architecture search (NAS) to the path-based knowledge graph embedding problem, designed the NAS search space and search algorithm for the problem, and demonstrated the necessity and effectiveness of the algorithm through experiments.

Strengths: Due to the unique input characteristics of entities and relationships of path-based knowledge graph embedding problem, a specific recurrent function is necessary, such as the manually designed function RSN. The author claims that specific structure needs to be designed for specific KG problem, so the NAS method is suitable in this scenario. The performance achieved by the NAS algorithm in Entity Alignment and Link Prediction tasks exceeds most benchmarks. It seems that the paper is the first study to apply the NAS method to Path-based knowledge graph embedding problems

Weaknesses: The weakness of one-shot NAS methods in complex search space is well-known, and common solution is to trade off computational complexity and performance. The algorithm proposed in this paper seeks balance by allocating different resources in different subspaces of the search space, and claims that it can be generalized to other complex NAS problems, but the trade-off of computational complexity and performance has been explored a lot, for example, arxiv.org/abs/1909.10815. The NAS algorithm in this paper is not very novel actually.

Correctness: The authors claim that the NAS algorithm can achieve better results in the path-based knowledge graph embedding problem makes sense. The study supports this claim in section4.2/4.3 empirically. What is the difference between S1 and S2 in Section 4.2? Why do p2 and p3 behave more differently on S1 and behave the same on S2? In the comparison of section4.4, for DARTS, the gradient is obtained from loss on training set since validation metric is not differentiable. Is it fair to be compared with the algorithm optimized based on validation metric?

Clarity: The writing of the article is good. There are a few errors, such as “uptimize” on line 179, figure 2 on line 206 which should be figure 3, and so on.

Relation to Prior Work: The article compares its search space with previous works such as PTransE, Chains, RSN, etc., and shows the benefits of NAS method. If the article claims that the proposed NAS algorithm is sufficiently innovative, it’s necessary to be compared with more benchmarks elaborately. In addition, the content of this paper is almost completely consistent with arxiv.org/abs/1911.07132.

Reproducibility: Yes

Additional Feedback:

[Author Response · NeurIPS 2020]

**To Reviewer#1 Q1.** *The method needs to search through paths for long-term information, it's like to find conflict facts or relations. How to deal with the conflict is not mentioned.* **Reply:** The conflict problem in BootEA [42] is by their bootstrapping training strategy, i.e., adding generated facts (not necessarily true) in training data. Same as baselines [11,19,46,49] and survey [48], we work on KG with standard training sets (i.e., with only true facts and no conflicts).

**Q2.** *It would be interesting to include 2 extra methods, SRAP with only macro and SRAP with only micro, to illustrate the effectiveness of macro/micro split.* **Reply:** All the architectures are defined by micro+macro part and thus cannot be evaluated separately. In Fig. 5, we have "search with only macro stage" (*Random*, *Reinforce*, *Bayes*) and "with only micro stage" (*ASNG*, *DARTS*). We will elaborate more on this in Sec. 4.4. Besides, in appendix Sec. B.2 (Fig. 8), we show that the split is reasonable.

**Q3.** *Some recent work on graph alignment were not included in the comparison.* **Reply:** For the three papers, the first one BootEA [42] and the third one GCN-Align [49] are included in our submission and are carefully compared in Sec. 4.3 (see Tab. 4). The code of VR-GNN [Ye et al. 2019] is not publicly available. Thus, we report the performance (H@1, H@10, MRR) by our implementation in the right table. As can be seen, SRAP is much better.

|  | DBP-WD | DBP-YG | EN-FR | EN-DE |
| --- | --- | --- | --- | --- |
| VR-GCN | 19.4 55.5 0.32 | 33.0 68.7 0.45 | 26.9 63.9 0.38 | 40.6 73.1 0.52 |
| SRAP | **40.7 71.2 0.51** | **40.2 72.0 0.51** | **35.5 67.9 0.46** | **50.1 75.6 0.59** |

**To Reviewer#2 Q1.** *The search cost still takes tens of hours. The search algorithm is conducted independently on each data set.* **Reply:** As in Tab. 6, the searching cost is already comparable with fine-tuning and SRAP already performs best. This is the best case that NAS can do [1,28,55]. Considering data similarity is a promising extension direction.

**Q2.** *It is not sure whether the search problem will be influenced when the paths change.* **Reply:** No. Paths just serve as the input to the search problem but will not change the problem definition.

**Minor.** *(i) natural policy gradient is costly.* **Re:** Unlike the single-level optimization [2,32], NG is used in the bi-level problem Eq.(2) and is cheaper compared with updating embeddings $F^*$. *(ii) It's better to include Path-ranking as a reference.* **Re:** Thanks for the suggestion to provide a good reference to support our motivation. *(iii) why not sample the path and train the model simultaneously?* **Re:** The sampling process is expensive [18,19] and the model already performs well when the paths are fixed. *(iv) If the graph evolves, can this algorithm adapt to the new one?* **Re:** Same as the baselines, SRAP is on the static KG. Adapting to evolving KG is a future direction. We will add above discussions.

**To Reviewer#3 Q1.** *It seems that the model does not do well in terms of efficiency. How does it scale to large datasets?* **Reply:** In Tab. 1, SRAP has the same complexity as path-based models [11,19,27]. In Tab. 6, we show the search cost is comparable with the fine-tune cost. These evidences show that efficiency is not an issue. We will add results on YAGO3-10 dataset [Mahdisoltani et al. 2015] (41.4 66.6 0.50 for SRAP in Tab. 5 within 100h) with millions of triplets.

**Minor.** *(i) What does the sentence "based on the relational paths ... individually" mean?* **Re:** It means that the relational paths can be the input of triplet-based models and these models work on each triplet in the paths separately. We will clarify these statements to be more precise. *(ii) CompGCN and AutoSF should be added.* **Re:** We will add the two works for comparison and discussion. The scalability of CompGCN is a problem due to the complexity $O(|\mathcal{S}|d)$, $|\mathcal{S}|$ is triplet number. AutoSF is a triplet-based model, which can be combined with SRAP to form a much larger search space. A combination of the searched model of AutoSF and SRAP on WN18RR gives 44.9, 55.8, 0.49 in Tab. 5, which can further improve upon SRAP. More results will be elaborated in the final version.

**To Reviewer#4 Q1.** *The trade-off of computational complexity and performance has been explored a lot, e.g. arxiv-1909.10815. The NAS algorithm in this paper is not very novel actually.* **Reply:** This paper Balanced-NAO [Luo et. al.] does not weaken the novelty of our hybrid-search algorithm. It supports us for the problem of one-shot methods. They give different resources to different models. The similar idea has been previously explored by Hyperband [Li et al. JMLR 2018]. Instead, we use micro/macro algorithms for different parameters in the search space. The two approaches are orthogonal and can be combined together. We will add more references for discussion.

**Q2.** *What is the difference between S1 and S2 in Section 4.2? Why do p2 and p3 behave more differently on S1 and behave the same on S2?* **Reply:** Indeed, the difference in AUC-PR score is small. We carefully check details in S1 and S2, and find that the queries in S1 has two support rules $neighbor \wedge locatedin \rightarrow locatedin$ and $locatedin \wedge locatedin \rightarrow locatedin$, whereas S2 only has the first one. P2 models on two triplets, and thus learns well in both S1 and S2. But for P3, the additional rule in S1 gives more paths and S3 may overfit on the training data. This slightly weakens the performance on S1 than S2. We will include this discussion in the final version.

**Q3.** *Is it fair to compare DARTS on training loss with the algorithm optimized based on validation metric?* **Reply:** Please note that optimizing architectures on the training set is also an alternative practice in the NAS literature, see SNAS [Xie et al. ICLR2019], Auto-DeepLab [Liu et al. CVPR2019], MiLeNAS [He et al. CVPR2020].

**Q4.** *If the article claims that the proposed NAS algorithm is sufficiently innovative, it's necessary to be compared with more benchmarks elaborately.* **Reply:** As stated in Q1, the algorithm is novel. Our main target is not to propose an universal NAS algorithm, but one can explore domain-information in KG well. Main baselines are compared with those in Tab. 4&5, which are benchmarks in the KG literature [19,41,48].

[Meta-Review · NeurIPS 2020]

The paper describes a recurrent neural architecture search technique to leverage path information in knowledge graphs. This is an important contribution both in terms of architecture design and in terms of improving the state of the art.